# The PSI Domain of the MET Oncogene Encodes a Functional Disulfide Isomerase Essential for the Maturation of the Receptor Precursor

**DOI:** 10.3390/ijms232012427

**Published:** 2022-10-17

**Authors:** Dogus Murat Altintas, Simona Gallo, Cristina Basilico, Marina Cerqua, Alessio Bocedi, Annapia Vitacolonna, Orsola Botti, Elena Casanova, Ilaria Rancati, Chiara Milanese, Sara Notari, Giorgia Gambardella, Giorgio Ricci, Pier Giorgio Mastroberardino, Carla Boccaccio, Tiziana Crepaldi, Paolo Maria Comoglio

**Affiliations:** 1IFOM, FIRC Institute for Molecular Oncology, Via Adamello 16, 20139 Milano, Italy; 2Department of Oncology, University of Turin, 10060 Candiolo, Italy; 3Candiolo Cancer Institute, FPO-IRCCS, Strada Provinciale 142, 10060 Candiolo, Italy; 4Department of Chemical Sciences and Technologies, University of Rome “Tor Vergata”, 00133 Rome, Italy; 5Department of Life, Health, and Environmental Sciences, University of L’Aquila, 67100 L’Aquila, Italy

**Keywords:** MET, oncogene, PSI domain, protein disulfide isomerase, protein maturation, Golgi apparatus

## Abstract

The tyrosine kinase receptor encoded by the MET oncogene has been extensively studied. Surprisingly, one extracellular domain, PSI, evolutionary conserved between plexins, semaphorins, and integrins, has no established function. The MET PSI sequence contains two CXXC motifs, usually found in protein disulfide isomerases (PDI). Using a scrambled oxidized RNAse enzymatic activity assay in vitro, we show, for the first time, that the MET extracellular domain displays disulfide isomerase activity, abolished by PSI domain antibodies. PSI domain deletion or mutations of CXXC sites to AXXA or SXXS result in a significant impairment of the cleavage of the MET 175 kDa precursor protein, abolishing the maturation of α and β chains, of, respectively, 50 kDa and 145 kDa, disulfide-linked. The uncleaved precursor is stuck in the Golgi apparatus and, interestingly, is constitutively phosphorylated. However, no signal transduction is observed as measured by AKT and MAPK phosphorylation. Consequently, biological responses to the MET ligand—hepatocyte growth factor (HGF)—such as growth and epithelial to mesenchymal transition, are hampered. These data show that the MET PSI domain is functional and is required for the maturation, surface expression, and biological functions of the MET oncogenic protein.

## 1. Introduction

The MET proto-oncogene encodes the tyrosine kinase receptor for hepatocyte growth factor(HGF). MET is an essential gene involved in organ development, angiogenesis, and wound healing [1,2,3]. In cancer, it acts as an oncogene, conferring on tumor cells the ability to survive and escape the hostile primary tumor microenvironment and form distal metastases [4]. MET is altered in multiple types of cancer and behaves as a pivotal regulator of invasive growth [5]. Indeed, MET was recently proposed to stand among the top five proteins to prioritize in targeted therapy [6]. Therefore, it is essential to fully understand MET biology, including the functions of all its domains.

MET is translated into a precursor protein of 175 kDa that matures through proteolytic cleavage by furin in the Golgi apparatus [7]. The mature protein is composed of a heterodimer of an extracellular α subunit (50 kDa) linked via disulfide bonds to the transmembrane β subunit (145 kDa) [8]. The extracellular portion of the MET β chain is composed of a SEMA domain, a plexin–semaphorin–integrin (PSI) homology domain, and four immunoglobulin-like IPT domains [4,9,10]. The juxtamembrane domain and the catalytic site comprise the intracellular portion (Figure 1A). Although the SEMA domain and the IPTs of the MET protein are well-characterized and are indispensable for HGF binding [11,12], only descriptive structural data about the MET PSI domain are available, proposing it as a linking module for proper ligand binding [13,14]. Biochemical functions of the PSI domain remain to be elucidated.

The MET PSI domain is a cysteine-rich conserved sequence with two non-canonical CXXC motifs observed in disulfide isomerases [15,16]. We demonstrate that the domain is a functional disulfide exchange isomerase required for proper folding and cleavage of the MET precursor and exposure of the mature receptor at the cell surface.

## 2. Results

### 2.1. The MET Extracellular Domain Displays Disulfide Isomerase Activity In Vitro

Sequence alignment analyses show that the PSI domain is highly conserved between MET, RON, integrins, PLEXINA1, and SEMA4D, which are all molecules involved in cell growth and invasion. Notably, seven cysteines and two CXXC motifs display complete conservation between the proteins mentioned above, highlighting their potential functional importance (Figure 1B). Indeed, in integrins, it was recently demonstrated that the PSI domain confers protein disulfide isomerase activity through the CXXC motifs [13]. 

We explored whether the MET PSI domain has a similar enzymatic property. We engineered and produced the extracellular domain of MET (MET ECD) and purified it by affinity chromatography. As anticipated, Coomassie stainings display a single 90 kDa β and 50 kDa α chains under reducing conditions, and a single 140 kDa α-β heterodimer under non-reducing conditions (Figure 2A). 

We incubated the MET ECD with a scrambled RNAse A and tested its ability to restore the native disulfide bonds using the traditional method of Kunitz [18]. Fluorescence quenching experiments show that MET ECD binds to sRNAse A with an affinity comparable to one of other human protein disulfide isomerases known to interact with sRNase A ([19], *K*_D_ ≈ 0.3 ± 0.1 µM, Figure 2B). We then sought to determine if this interaction is followed by a possible disulfide isomerase activity of MET ECD. We observe that MET ECD is able to restore RNAse A activity (Figure 2C), signifying that MET ECD has protein disulfide isomerase activity within the range of the one measured with integrin *α*2/β3 (ITGA2B3) and the isolated PSI domain of integrin β3 (ITGB3 PSI, [13], Figure 2C). All tests were run in the presence of EDTA, ruling out any possible involvement of divalent cations [20]. We then tested the inhibitory activity of the DO24 antibody (Fab fragment), which is known to bind the MET PSI domain [21]. The antibody fully inhibits the MET ECD disulfide isomerase activity (Figure 2D). Another MET antibody, DN30 (Fab fragment), directed against the IPT-4 domain [22], behaves as a negative control (Figure 2D).

Collectively, these results suggest that MET PSI domain is a functional disulfide isomerase (Figure 2E).

### 2.2. The PSI Domain Is Crucial for MET Maturation

These data prompted us to study the effect of disulfide isomerase activity on MET biology. We treated EBC1 cells endogenously expressing wild-type MET with a known PDI inhibitor, 16F16 [23,24,25]. Pharmacological inhibition leads to a significant accumulation of the MET 175 kDa precursor (Figure 3A), suggesting that *α*/β cleavage is influenced by cellular PDI activity. 

Next, we verified the involvement of the MET PSI domain in cellular PDI activity by determining the cellular thiol–disulfide redox equilibrium [26,27]. TOV112D cells expressing wt MET or MET∆PSI were lysed in the presence (recovering membrane proteins) or in the absence of SDS (recovering soluble proteins). Reduced thiols were labelled with maleimide conjugated to Alexa Fluor 647 (green pseudocolor), and the oxidized thiols were labelled using maleimide conjugated to Cy3 (red pseudocolor). As exhibited in Figure 3B, the MET PSI domain profoundly affects the reduced and oxidized thiol levels, highlighting its critical involvement in disulfide bridge formation through its disulfide isomerase activity.

To assess if the MET maturation is regulated by its endogenous disulfide isomerase activity, we deleted the whole PSI domain (MET_∆PSI) and transfected it in MET-negative TOV112D cells. wt_MET constructs were used as control. The proteins translated from wt_MET is resolved in two bands, a major corresponding to the mature MET β chain (145 kDa) and a minor corresponding to the precursor (175 kDa). By contrast, in MET_∆PSI-transfected cells, the relative ratio between the two bands is inverted, suggesting that the cleavage is hampered (Figure 3C). Similarly, mutations of the first (C529XXC532), the second (C538XXC541), or both motifs to AXXA lead to the accumulation of the precursor (Figure 3C). Superimposable results were obtained using mutations of the first or both CXXC motifs to SXXS or AXXA and transient transfection in HEK293T cells (Figure 3D,E), showing that these motifs have a pivotal role in MET maturation.

### 2.3. MET_CXXC Mutants Are Trapped in the Golgi Apparatus and Constitutively Phosphorylated

We next investigated the subcellular localization of the MET precursor in CXXC mutants. As expected [28], confocal immunofluorescence experiments show that the MET signal is localized in cells expressing the wt receptor in the Golgi apparatus and the plasma membrane. In CXXC mutants, MET is trapped in the Golgi (Figure 4A). Similarly, flow cytometry experiments show that the amount of membrane MET is significantly reduced in mutants compared to wt (Figure 4B). Comparable results were obtained in transiently transfected HEK293T cells (Figure 4C). 

Since the precursor accumulates, we investigated if tyrosine kinase is constitutively phosphorylated by clustering in the Golgi. To avoid constitutive phosphorylation due to uncontrolled overexpression and clustering of the kinase [29] in transfected cells, we infected TOV112D cells with lentiviral particles at a multiplicity of infection inferior or equal to one and established stable clones. Figure 5A shows that the wt_MET-expressing cells treated with HGF display phosphorylated MET in the membrane, while in CXXC mutants, the Golgi-gathered precursor is constitutively phosphorylated. The HGF-independent activation of the CXXC mutants is confirmed by Western blot (Figure 5B) and by flow cytometry in permeabilized cells (Figure 5C). 

### 2.4. The Misplaced MET Is Biologically Inactive

MET activates multiple signal transducers, including PI3K/AKT (survival/migration) and MAPK (proliferation). The subsequent activation of those pathways results in epithelial–mesenchymal transition (EMT) [30,31]. We, therefore, verified the activation status of the targets mentioned above and the occurrence of EMT. Despite the constitutive phosphorylation in the Golgi of CXXC mutants, only weak AKT or MAPK activations are observed (Figure 6A). Accordingly, in the presence of HGF, the viability of cells expressing wt MET is superior to cells expressing CXXC mutants (Figure 6B). EMT, assessed by Vimentin quantification by flow cytometry, is obviated in cells expressing CXXC mutants (Figure 6C). The expression of genes associated with the mesenchymal state is not induced in CXXC mutants (Figure 6D). As expected, control cells expressing wt MET respond to HGF.

## 3. Materials and Methods

### 3.1. Chemicals and Antibodies

Ribonuclease A (RNAse) from bovine pancreas (Type XII-A, 75–125 Kunitz units/mg protein), dithiothreitol (DTT), ethylendiamminotetreaacetic acid (EDTA), 5,5′-dithiobis(2-nitrobenzoic acid) (DTNB), L-glutathione (GSH), oxidized glutathione (GSSG), and ribonucleic acid (RNA) from baker’s yeast (*S. cerevisiae*) were purchased from Sigma-Aldrich (St. Louis, MO, USA). Sarcosine was purchased from Thermo Scientific™ (Waltham, MA, USA). HGF was purchased from R&D (Minneapolis, MN, USA), and 16F16 was purchased from Selleck Chemicals LLC (Houston, TX, USA). All antibodies used in this study are listed in Appendix A.

### 3.2. Generation, Expression and Purification of the MET Ectodomain

cDNA sequences of human MET ectodomain (MET ECD: Δ25–932) and mouse β3 integrin PSI (1–56) were C-terminally tagged with Flag-His. The engineered soluble receptor was produced by transfection of HEK293T cells and purified from cell supernatants by affinity chromatography using HisTrap HP columns (GE Healthcare, Freiburg, Germany) according to the manufacturer’s instructions. Large-scale protein production and purification were performed by U-Protein Express BV (Utrecht, The Netherlands). Human α2β3 integrin was purchased from R&D (Minneapolis, MN, USA).

### 3.3. Preparation of Scrambled RNAse and RNAse Activity Assay

The scrambled RNAse was prepared as previously described [5]. Titration with DTNB confirmed the complete reduction of the four disulfides. The purity of the sRNAse sample was analyzed by SDS-PAGE. Activities of native RNAse and sRNAse were assayed by the ribonuclease A detection kit (Sigma-Aldrich, St. Louis, MO, USA) as previously described [18]. All the solutions tested were incubated at 30 °C in 5 mM EDTA, 50 mM sodium phosphate buffer pH 7.5. At different times, an aliquot was taken and added to a solution containing RNA (0.05% *w*/*v*) in 50 mM sodium acetate buffer pH 5.0. Restored activities were monitored with a spectrophotometer at 25 °C with λ = 300 nm. Since different batches of sRNAse could give slightly different results [32], analyses were carried out using the same batch of sRNAse in all the experiments.

### 3.4. Fluorescence Measurements

The fluorescence analysis of MET ECD was performed on a FluoroMax-4 Horiba spectrofluorometer (Kyoto, Japan) with a quartz cuvette of 1 × 0.4 cm path length. Quenching of the intrinsic fluorescence was used to calculate the *K*_D_ value for the interaction between MET ECD and sRNase [33]. The set panel for determining *K*_D_ was slits of 5–8 nm, excitation wavelength of 295 nm, and emission spectra of 300–400 nm. The spectra of MET ECD (0.51 µM) in 50 mM sodium phosphate buffer, pH 7.5, were recorded in the presence of different sRNase concentrations (from 0.1 to 2 µM) at 25 °C. The maximum intensities of each spectrum (351 nm) were subtracted from the maximum intensities (361 nm) acquired for NATA in the same conditions.

### 3.5. Cell Culture

The human ovarian carcinoma TOV112D (RRID:CVCL_3612, ATCC CRL-11731™), the human lung squamous cell EBC1 (RRID:CVCL_2891, Creative Bioarray CSC-C6336J), and the human embryonic kidney HEK293T (RRID:CVCL_0063, ATCC CRL-1573™) cell lines were purchased and cultured according to manufacturer’s instructions in a humidified atmosphere at 37 °C. Cells were passed and subcultured to ~80/90% of confluence and were regularly verified for the absence of mycoplasma contamination.

### 3.6. Lentiviral Vectors

The lentiviral plasmid containing the complete cDNA of the human wild-type MET was a gift from David Rimm (Addgene plasmid #37560; http://n2t.net/addgene:37560; RRID:Addgene_37560; accessed on 12 December 2021). The MET PSI domain complete deletion and amino acid substitutions were performed using the QuickChange II Site-Directed Mutagenesis Kit (Agilent Technologies, Santa Clara, CA, USA) using primers listed in Appendix A. The indicated vectors were transfected into TOV112D cells using Lipofectamine 2000 reagent (Thermo Scientific™, Waltham, MA, USA). TOV112D-derived stable clones were produced by lentiviral infections utilizing a multiplicity of infection (MOI) inferior or equal to 1 to avoid MET overexpression. Cells were selected in the culture medium containing 1.5 μg/mL of puromycin for two weeks and sorted by flow cytometry using the GFP marker. The expression of the MET protein in each clone was verified by Western blot.

### 3.7. Plasmids

Codon optimization, synthesis of MET (GenBank: NM_000245.4), and expressed mutagenesis for SXXS mutants were commercially performed by GenScript (Piscataway, NJ, USA) in the pcDNA3.1(+)-myc-His A plasmid. Plasmids were used for transient transfection of HEK293T cells using Lipofectamine 2000 reagent (Thermo Scientific™, Waltham, MA, USA).

### 3.8. Western Blot Analyses

Cells were lysed in Laemmli buffer as previously described [34]. Proteins were separated in denaturing polyacrylamide gels and blotted against antibodies listed in Appendix A. The Precision Plus Protein Dual Color Standards (Bio-Rad Laboratories S.r.l., Hercules, CA, USA) were used as a ladder to verify protein size. 

### 3.9. Flow Cytometry

Formaldehyde-fixed cells were permeabilized with Triton X-100 (intracellular FACS) or not (membrane proteins), washed with PBS-1%BSA and incubated with antibodies listed in Appendix A. Samples were analyzed on a CyAn ADP LX nine-color analyzer (Beckman Coulter, Brea, CA, USA). 

### 3.10. Immunofluorescence Analysis

TOV112D cells were plated in 8 wells Nunc™ Lab-Tek II™ Chamber Slides™. Immunofluorescence experiments were carried out as previously described [35]. Images were acquired by the Leica TCS SP2 AOBS confocal laser-scanning microscope (Leica Microsystems). 

### 3.11. Viability Assay

Cells were plated in 96-well plates at a density of 5000 cells/well. After 24 h starvation, cells were treated with HGF. Cell viability was assessed using CellTiter 96^®^ Non-Radioactive Cell Proliferation Assay (Promega, G4100) for 48 h. Results are mean +/− SEM of N = 6 replicates.

### 3.12. Statistical Analyses

Data are reported as mean ± SEM of N ≥ 3 independent experiments. ANOVA with Tukey’s post hoc test was performed to compare different groups with ns: not-significant; *: *p* < 0.05; **: *p* < 0.01; ***: *p* < 0.001; ****: *p* < 0.0001. 

### 3.13. Determination of Cellular Thiol–Disulfide Redox Equilibrium

Assays were performed as previously described, according to a procedure we refer to as redox immunohistochemistry [26,27]. Briefly, equal amounts of cells were harvested and suspended in a lysis solution containing Tris-HCl 50 mM, SDS 1%, EDTA 1 mM, NEM 10 mM (N-Ethylmaleimide, Sigma-Aldrich, St. Louis, Mo, USA, E3876), Alexa Fluor 647 Maleimide 20 uM (Thermo Scientific™, Waltham, MA, USA, A20347), and proteinase inhibitors mix to label reduced thiols. Lysates were incubated for 5 min at 70 °C, sonicated, and finally incubated for 30 min at RT in the dark. Quenching of unreacted thiols was performed with 100 mM NEM for 30 min. Protein extracts were then precipitated with a cold precipitation solution (50% acetone, 25% methanol, and 25% ethanol). Next, disulfide bonds were reduced with 20 mM TCEP (Tris(2-carboxyethyl)phosphine, Thermo Scientific™, Waltham, MA, USA, 20490). Newly reduced thiols were labelled with AlexaFluor 555 Maleimide 20 uM (Thermo Scientific™, Waltham, MA, USA, A20346). After precipitation, samples were diluted in Laemmli sample buffer, and proteins were resolved under reducing conditions. After electrophoresis, the gel was fixed with 5orthophosphoric 2.5% orthophosphoric acid overnight under mild agitation. Final images and analysis were performed with the Chemidoc MP Imaging System (Bio-Rad Laboratories S.r.l., Hercules, CA, USA).

## 4. Discussion

MET is an oncogene encoding the tyrosine kinase receptor of the hepatocyte growth factor. It is translated as a single chain 175 kDa precursor protein, cleaved in the Golgi apparatus into α (50 kDa) and β (145 kDa) chains by the protease furin [8]. Three interchain disulfide bonds link the heterodimer. Four intrachain bonds stabilize the three-dimensional conformation of the extracellular moiety (Figure 1A). In general, disulfide bond-rich proteins require remodelling by the chaperone action of protein disulfide isomerases (PDIs) [36]. The mechanisms of the Cys-rich MET precursor maturation are only partially understood.

MET contains a highly conserved extracellular domain (plexin–semaphorin–integrin homology domain, PSI), whose function is still unexplored. Notably, this domain contains two conserved CXXC motifs found in the catalytic site of PDIs [15,16]. This paper shows that the PSI domain has endogenous disulfide isomerase activity. This conclusion is based on the ability of the recombinant purified MET extracellular domain to restore the RNAse activity of a scrambled RNAse [18]. Moreover, monoclonal antibodies directed against the PSI domain inhibit the reaction in vitro. 

Deletion of the PSI domain, or site-directed mutations in CXXC motifs, hampers the cleavage of the single-chain MET precursor. The uncleaved precursor is trapped in the Golgi apparatus. As demonstrated in furin-negative cells, the lack of cleavage per se is not essential for MET translocation to the plasma membrane [7,37]. Thus, the precursor is stuck in the Golgi due to the lack of disulfide remodelling. The inefficiency of MET trafficking does not interfere with the autophosphorylation of the precursor, the latter being even constitutively active, probably due to protein clustering [29,38,39,40]. However, we observe that this phosphorylation does not lead to signal transduction. The intracellular signalling after MET activation is observed in the endosomal compartment and the nucleus [41,42,43,44,45]. We hypothesize that the disulfide bonds’ rearrangement to reach the protein’s proper conformation is required for signal transduction. However, we cannot rule out that the receptor’s misplacement might also be responsible for the absence of signalling, because MET docking proteins are located under the plasma membrane. Nevertheless, trapping MET into the Golgi apparatus by inhibiting disulfide isomerase activity through CXXC mutations impedes AKT and MAPK activation and epithelial to mesenchymal transition. We, therefore, demonstrate the pivotal role of the catalytic MET PSI domain in MET maturation, trafficking, and signalling. While the function of this domain is mainly in the Golgi apparatus in “*cis*”, we cannot exclude the possible function(s) in “*trans*”. Indeed, we reveal that the PSI domain profoundly affects the cellular thiol–redox equilibrium. Therefore, the MET PSI domain may serve as a chaperone for the folding of other proteins. In addition, MET PSI domain is exposed at the cell surface. It may act as a disulfide isomerase towards other cell surface proteins or components of the extracellular matrix. Mass-spectrometry-based approaches could identify such potential substrates and open new insights into MET signalling and, subsequently, tumor biology. 

Recently, it was established that the inhibition of integrin PDI activity reduced platelet aggregation in vitro and thrombus formation in vivo [19,46,47,48,49]. Targeting the general PDI activity was proposed as a new approach to fight against diseases related to intravascular platelet aggregation [49]. However, the widespread distribution of PDI in the body might result in severe toxicity [50,51,52]. Hitherto, targeting specifically the MET disulfide isomerase activity in a therapeutic window can lead to the impairment of MET functions and tumor growth. Likewise, an antibody targeting the MET PSI domain hampered MET signalling in vitro and reduced tumor growth in preclinical models [53]. We, thus, reflect that further studies aiming to target MET disulfide isomerase activity can lead to the development of original therapeutic strategies.

## Figures and Tables

**Figure 1 ijms-23-12427-f001:**
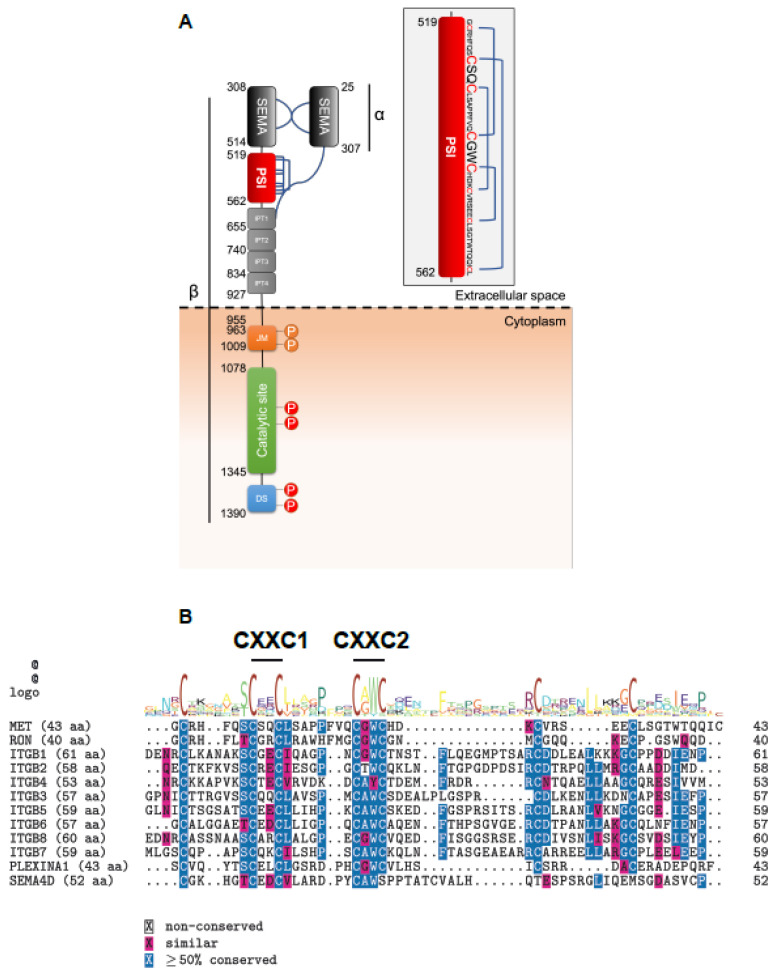
MET holds a Cys-rich, conserved PSI domain. (**A**) Schematics of MET. Interchain and intrachain disulfide bonds are presented in blue according to [1,2,3]. The PSI domain is exhibited as an insert, with cysteine residues in red and CXXC motifs in more prominent points. (**B**) PSI domain amino acid sequence alignment between the indicated proteins. Analyses were performed using the R package “msa” version 3.14 using the ClustalW algorithm [17]. Conserved amino acids are highlighted in blue, and biochemically similar amino acids are displayed in pink. The two CXXC motifs are shown above the consensus sequence.

**Figure 2 ijms-23-12427-f002:**
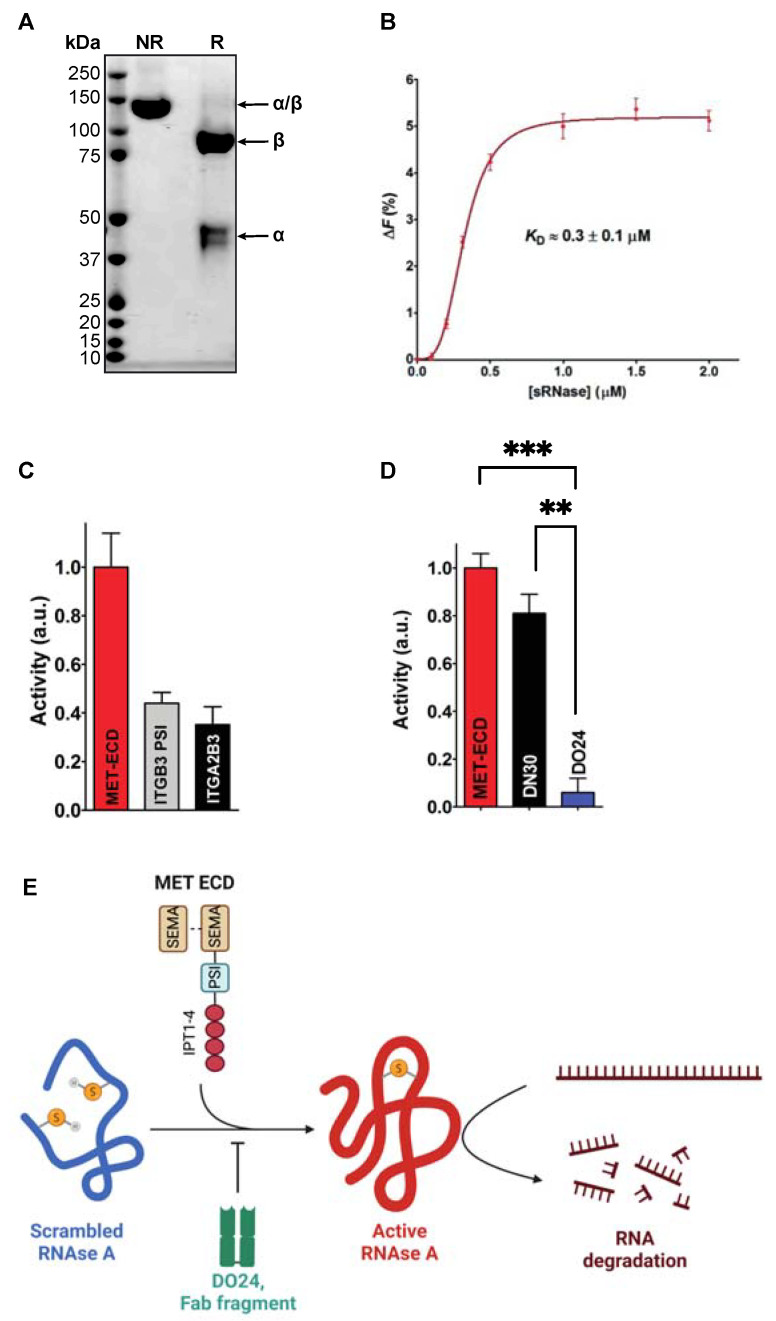
Biochemical characterization of the MET ECD protein disulfide activity. (**A**) Coomassie staining of the purified recombinant human MET ECD under non-reducing (NR) and reducing conditions (R). (**B**) Quenching of the intrinsic fluorescence of MET ECD (0.51 µM) after the addition of sRNase (from 0.1 to 2 µM). (**C**) Disulfide isomerase activity of MET ECD, mouse PSI domain of ITGB3, and ITGA2B3 were measured as described under Materials and Methods. Activity recovered by sRNase (2 µM) incubated with MET ECD (red bar), PSI domain of ITGB3 (grey bar), and ITGA2B3 (black bar). (**D**) Effect of antibodies on MET ECD disulfide isomerase activity. Activity recovered by sRNase (2 µM) incubated with MET ECD (red bar), MET ECD with anti-IPT1 domain (DN30) (black bar), and MET ECD with anti-PSI domain (DO24) (blue bar) as described in Materials and Methods. The error bars represent SEM from three independent experiments. ANOVA with Tukey’s post hoc test was performed to compare different groups with ns: not-significant; **: *p* < 0.01; ***: *p* < 0.001 (**E**) Schematics of the RNAse activity assay (modified from Kunitz [18]). The scrambled—inactive—RNAse A was incubated with the extracellular domain of MET (MET ECD). RNAse, thus, becomes active through the disulfide isomerase activity of MET ECD. This reaction is inhibited in the presence of DO24, an antibody binding to the MET PSI domain [21]. Created with BioRender.com.

**Figure 3 ijms-23-12427-f003:**
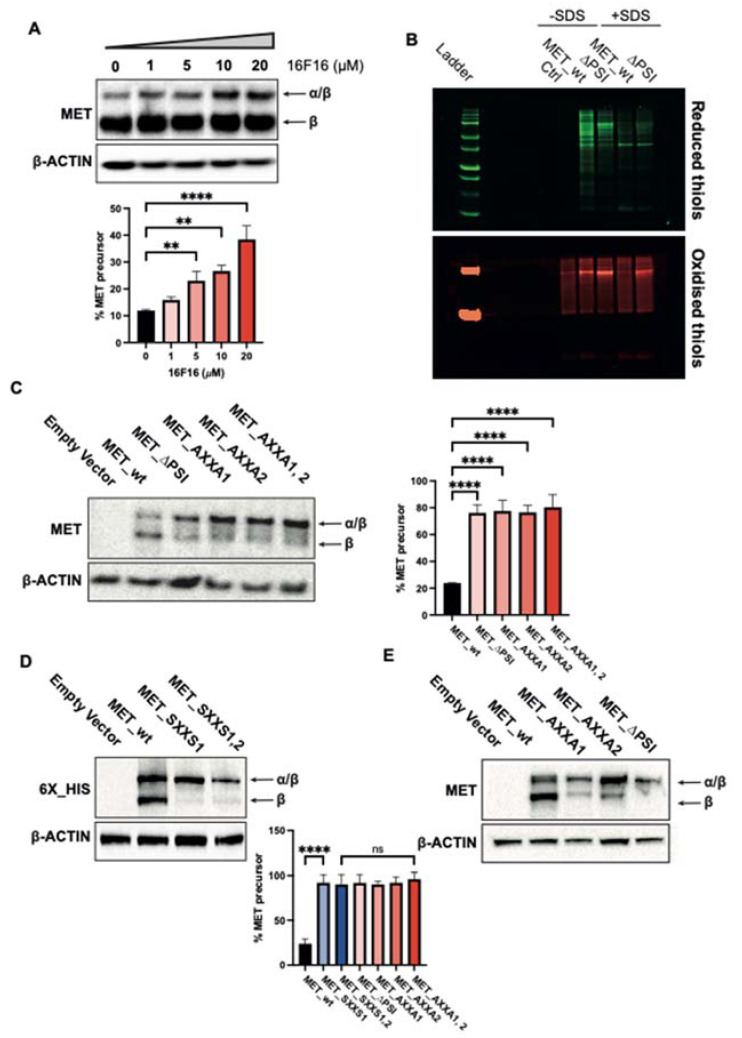
The MET PSI domain is indispensable for MET maturation through the CXXC motifs. (**A**) EBC1 cells were treated with indicated concentrations of 16F16 for 48 h or DMSO as vehicle. Proteins were resolved in 6% SDS-PAGE, and MET expression was assessed by Western blot. (**B**) TOV112D cells stably expressing MET wt or MET∆PSI were lysed and proteins’ thiol groups were labelled according to their redox state as described in the Section 3. Ctrl condition corresponds to cells labelled with Cy3-conjugated maleimide only to remove autofluorescence in the 647 channel. Reduced thiols are presented with the green pseudocolor and oxidized thiols are shown with the red pseudocolor. TOV112D (**C**) and HEK293T (**D**,**E**) cells were transiently transfected with indicated constructs, and cells were lysed 48 h post-transfection to assess MET expression with MET and 6X_HIS tag antibodies. Β-ACTIN was used as a loading control. Western blots were quantified using ImageJ software v2.1.0/1.53c. Results are mean +/− SEM with N = 3. ANOVA with Tukey’s post hoc test was performed to compare different groups with ns: not-significant; **: *p* < 0.01; ****: *p* < 0.0001.

**Figure 4 ijms-23-12427-f004:**
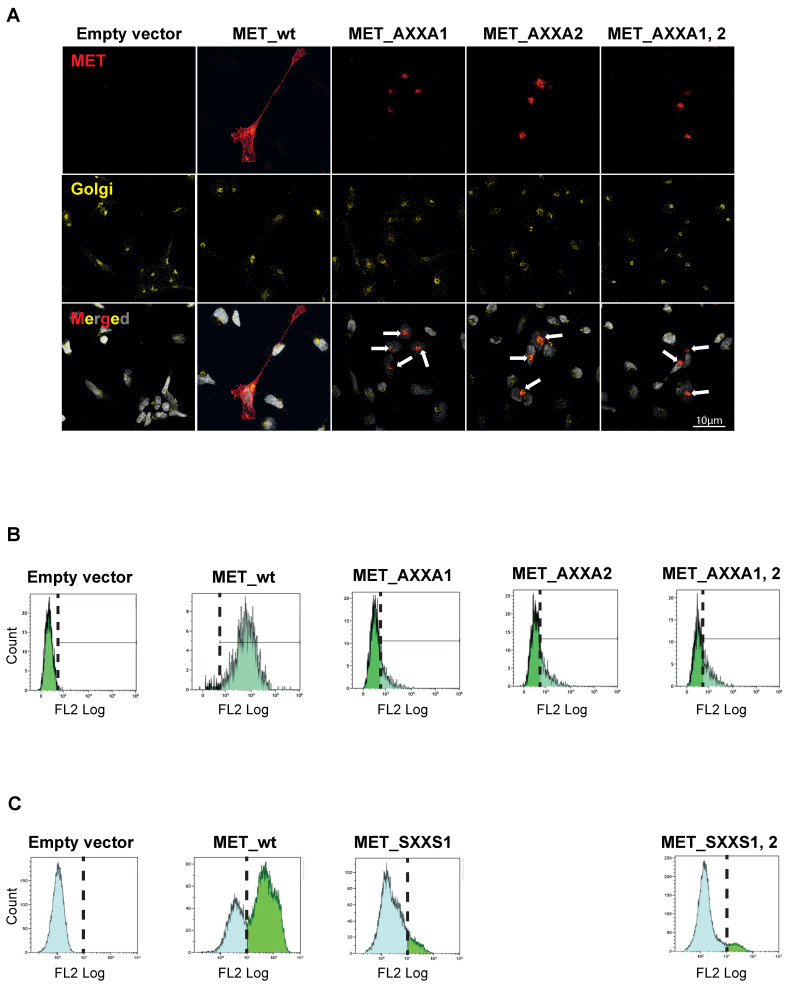
The MET CXXC mutants are blocked in the Golgi apparatus. (**A**) TOV112D cells stably expressing indicated receptors were stained with MET antibodies (red pseudocolor) and 58 K Golgi protein antibodies (yellow pseudocolor). DAPI staining was performed to visualize the nuclei (grey). Images were acquired with the Leica TCS SP8 confocal microscope and post-processed with ImageJ. Pictures are representative of two independent experiments. Arrows in the merged images denote Golgi gathering of the signal. (**B**) TOV112D cells stably expressing wt_MET or AXXA mutants of MET in indicated CXXC motifs were assessed for surface MET expression by flow cytometry. The dashed line indicates the background signal threshold determined in MET negative TOV112D cells. (**C**) HEK293T cells were transiently transfected with wt_MET or SXXS mutants and analyzed as above. In the latter cell line, the MET antibody recognizes both the endogenous (left blue peak superimposable to the peak observed in cells transfected with an empty vector) and transfected MET wild-type or mutant proteins (right green peak). The two peaks are separated with dashed lines.

**Figure 5 ijms-23-12427-f005:**
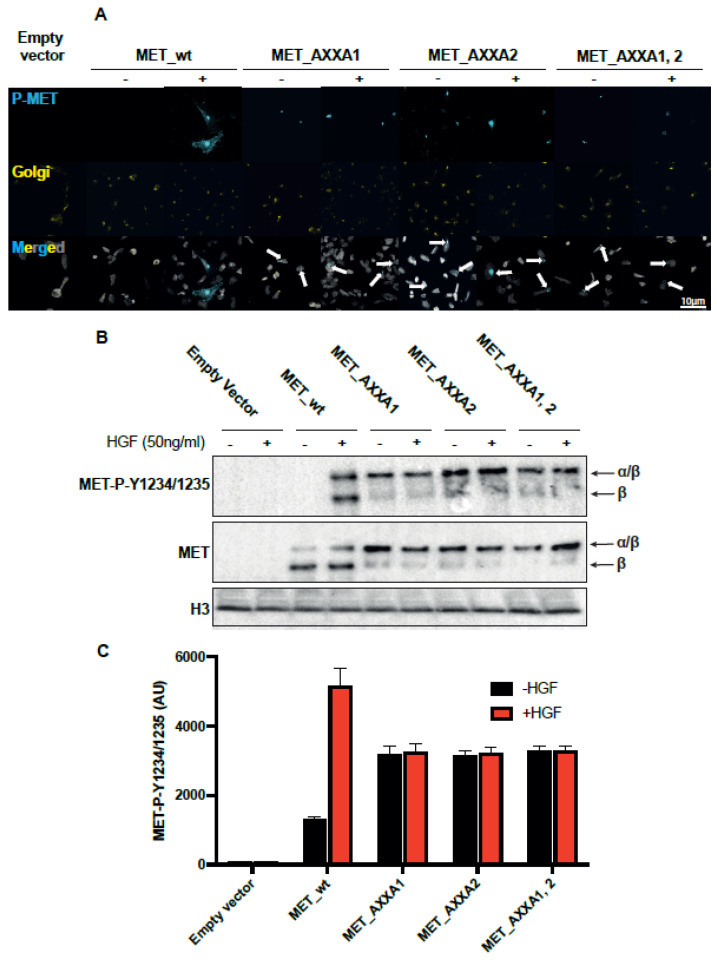
The MET CXXC mutants are constitutively phosphorylated. (**A**) TOV112D cells stably expressing indicated receptors were treated with ±50 ng/ml HGF for 15 min and stained with MET-P-Y1234/1235 antibodies (cyan pseudocolor) and 58 K Golgi protein antibodies (yellow pseudocolor). DAPI staining was performed to visualize the nuclei (grey). Images were acquired with the Leica TCS SP8 confocal microscope and post-processed with ImageJ. Pictures are representative of two independent experiments. Arrows in the merged images denote Golgi gathering of the P-MET signal. (**B**,**C**) Stable TOV112D clones expressing either MET_wt or the indicated mutant forms of MET were serum-starved for 24 h and treated with 50 ng/ml of HGF for 15’. Proteins were resolved in 6% SDS-PAGE and MET expression, and MET-P-Y1234/1235 levels were assessed by Western blot (**B**), and the amount of MET-P-Y1234/1235 was quantified by flow cytometry in permeabilized cells (**C**). Results are mean +/− SEM with N = 3.

**Figure 6 ijms-23-12427-f006:**
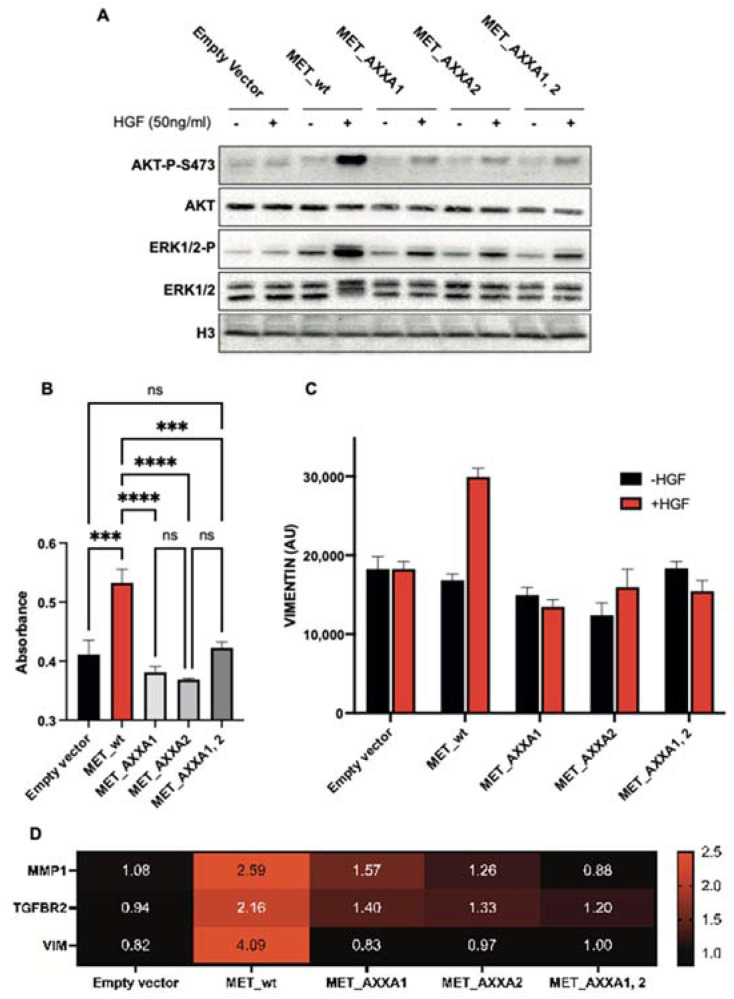
MET CXXC mutants are biologically inactive. (**A**) Stable TOV112D cells expressing either the wt_MET or indicated AXXA mutants were treated with HGF. Proteins were resolved in 10–12% gels to assess the activation levels of MET transducers, P-AKT and P-ERK1/2. Histone3 (H3) was used as a loading control. (**B**) Viability assay in the presence of HGF in indicated clones, measured by absorbance at 570 nm. Results are mean +/− SEM with N = 6. ANOVA with Tukey’s post hoc test was performed to compare different groups with ns: not-significant; ***: *p* < 0.001; ****: *p* < 0.0001. (**C**) Flow cytometry data in permeabilized cells assessing the amount of vimentin protein expression in wt_MET or CXXC mutants treated with 50 ng/ml of HGF for 48 h. Results are mean +/− SEM with N = 2. (**D**) Cells were treated with HGF for 6 h, and mesenchymal-state-associated gene expression was assayed by RT-qPCR. The geometric mean of TBP and 18S gene expression was used for normalization. Results are presented as heatmap of gene expression relative to -HGF condition.

## Data Availability

All raw data and non-commercial materials used in this study are available from the corresponding author upon reasonable request.

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
