# Peer review of "The PSI Domain of the MET Oncogene Encodes a Functional Disulfide Isomerase Essential for the Maturation of the Receptor Precursor"

_ijms, 2022, doi:10.3390/ijms232012427_

Round 1

Reviewer 1 Report

In this study, Altintas et al described a disulfide isomerase activity of the PSI extracellular domain of the MET receptor. PSI deletion or mutation of the two conserved CXXC motifs leads to defect of MET maturation in alpha and beta chain and accumulation of the MET immature form in Golgi. While these immature mutated forms of MET display constitutive phosphorylation, they were unable to induce activation of downstream signaling pathways and biological responses including EMT.

Although the observation that PSI domain could displays disulfide isomerase activity is very interesting and unexpected, some important experiments should be performed to support the finding and improve the study.

1.       Disulfide isomerase activity was evidenced only with one technique with purified MET-ECD and based on restoration activity of RNAse. Because demonstration of MET PSI domain disulfide isomerase activity is an important step, this should be confirmed with alternative methods.

2.      Figure 2: recombinant MET-ECD is able to restore RNase activity. It would be important, according to the identification CXXC motif in the figure 3, to evaluate disulfide isomerase activity of recombinant proteins mutated on these motifs.

3.      If PSI domain display disulfide isomerase activity, MET disulfide bridges could be affected upon PSI deletion or CXXC motif mutation. Therefore, disulfide bridge of MET WT and mutated should be assessed. This could be performed by MS.

4.      Furthermore, cysteine residues of CXXC motifs are directly involved in disulfide bridges. Therefore, CXXC mutations potentially lead to both disruption of disulfide isomerase activity and specific disulfide bridge involving the mutated cysteine. Analyses of disulfide bridges by MS could assess whether CXXC motif mutations lead to disruption of all the disulfide bridges, suggesting a disulfide isomerase activity supported by the motif, or only disruption of the bridges involving the mutated cysteines.

5.      Accumulation of the immature form for the MET CXXC mutant suggests a defect of the furin cleavage. The functional link between the cleavage defect and potential defect of the disulfide bridge should be investigated.

6.      Accumulation of immature MET in the Western blot shown in figure 3A, B and C should be quantified.

7.      In TOV cells and HEK cells two different mutated cells were used (AXXA or SXXS mutations). The choice of these different mutations should be justified. At least both mutants should be transfected in HEK.

8.      In the figure 4A, a higher magnification should be presented to see membrane and Golgi localization.

9.      Figure 4B: Similar experiments could be performed with permeabilized cells (as presented in figure 5A).

10.  Figure 5A: A full MET staining could be done.

11.  Figure 6A: Reprobe with MET and P-MET should be shown.

12.  Figure 6B: HGF stimulation promoted cell viability in MET WT expressing cells. This suggests either that this cell line display basal cell death counteracted by HGF or that HGF is able to favor cell proliferation. Specific evaluation of cell death and/or cell proliferation should be performed.

Reviewer 2 Report

The work was performed with mean level of quality. Some improvements are needed in  the methods section including all approaches with detail on statistical analyses. Images can be updated to better explain the insights. Editing is necessary. The idea and goal boost the clinical significance and open the discussion to the what's next in this specific field.

Round 2

Reviewer 1 Report

In this revised version, the authors responded properly to all my comments.